# Effect of Oral Nutritional Supplements with Sucromalt and Isomaltulose versus Standard Formula on Glycaemic Index, Entero-Insular Axis Peptides and Subjective Appetite in Patients with Type 2 Diabetes: A Randomised Cross-Over Study

**DOI:** 10.3390/nu11071477

**Published:** 2019-06-28

**Authors:** Lisse Angarita Dávila, Valmore Bermúdez, Daniel Aparicio, Virginia Céspedes, Ma. Cristina Escobar, Samuel Durán-Agüero, Silvana Cisternas, Jorge de Assis Costa, Diana Rojas-Gómez, Nadia Reyna, Jose López-Miranda

**Affiliations:** 1Escuela de Nutrición y Dietética, Facultad de Medicina, Universidad Andres Bello, Sede Concepción 4260000, Chile; 2Facultad de Ciencias de la Salud, Universidad Simón Bolívar, Barranquilla 080003, Colombia; 3Centro de Investigaciones Endocrino-Metabólicas “Dr. Félix Gómez”, Escuela de Medicina. Facultad de Medicina, Universidad del Zulia, Maracaibo 4001, Venezuela; 4Departamento de Medicina Física y Rehabilitación, Hospital “12 de Octubre”, Madrid 28041, Spain; 5Escuela de Nutrición y Dietética, Facultad de Ciencias para el Cuidado de la Salud, Universidad San Sebastián, Santiago 7500000, Chile; 6Escuela de Salud, Universidad Tecnológica de Chile, INACAP, Sede Concepción, Talcahuano 4260000, Chile; 7Faculty of Medicine/UniFAGOC, Ubá 36506-022, Minas Gerais, Brazil; 8Universidade do Estado de Minas Gerais (UEMG), Barbacena 36202-284, Minas Gerais, Brazil; 9Escuela de Nutrición y Dietética, Facultad de Medicina, Universidad Andres Bello, Santiago 8370321, Chile; 10Lipids and Atherosclerosis Unit, Maimonides Institute for Biomedical Research in Cordoba, Reina Sofia University Hospital, University of Córdoba, 14004 Córdoba, Spain; 11CIBER Physiopathology of Obesity and Nutrition (CIBEROBN), Institute of Health Carlos III, 28029 Madrid, Spain

**Keywords:** glycaemic index, incretins, subjective appetite, isomaltulose, sucromalt, nutritional supplement

## Abstract

Oral diabetes-specific nutritional supplements (ONS-D) induce favourable postprandial responses in subjects with type 2 diabetes (DM2), but they have not been correlated yet with incretin release and subjective appetite (SA). This randomised, double-blind, cross-over study compared postprandial effects of ONS-D with isomaltulose and sucromalt versus standard formula (ET) on glycaemic index (GI), insulin, glucose-dependent insulinotropic polypeptide (GIP), glucagon-like peptide 1 (GLP-1) and SA in 16 individuals with DM2. After overnight fasting, subjects consumed a portion of supplements containing 25 g of carbohydrates or reference food. Blood samples were collected at baseline and at 30, 60, 90, 120, 150 and 180 min; and SA sensations were assessed by a visual analogue scale on separate days. Glycaemic index values were low for ONS-D and intermediate for ET (*p* < 0.001). The insulin area under the curve (AUC_0–180 min_) (*p* < 0.02) and GIP AUC (*p* < 0.02) were lower after ONS-D and higher GLP-1 AUC when compared with ET (*p* < 0.05). Subjective appetite AUC was greater after ET than ONS-D (*p* < 0.05). Interactions between hormones, hunger, fullness and GI were found, but not within the ratings of SA; isomaltulose and sucromalt may have influenced these factors.

## 1. Introduction

Diabetes mellitus (DM) is a complex metabolic disorder associated with long-term complications as a result of the interplay of genetic, epigenetic, environmental and lifestyle factors [1]. Nowadays, DM is considered a pandemic health problem and one of the top 10 killers, responsible for 1.6 million deaths in 2016 [2]. According to International Diabetes Federation projections, by 2045, 629 million people will be afflicted by DM, exhibiting the fastest rising prevalence of this phenomenon in the history of humanity, which the highest prevalence rates in North America and the Caribbean [3]. Thus, the epidemiological impact of this disease is translated into higher public health expenditures worldwide [4,5].

One of the most important strategies for the prevention and treatment of DM has been correct management of carbohydrate consumption, having been reviewed in dietary guidelines and recommendations stated by many scientific organisations worldwide [6]. The American Diabetes Association (ADA) has highlighted this need, considering nutritional therapy as the fundamental basis of glycaemic control in DM patients [7]. Besides that, the European Association for the Study of Diabetes (EASD) has also focused its recommendations on the amount and type of carbohydrates consumed [8], generating considerable interest in low glycaemic index (LGI) food prescription for management of DM2 [9]. Indeed, a recent international expert consensus debated about the clinical role of GI and glycaemic load (GL) in DM management [10], concluding that low GI and low GL diets have been associated with a reduction in the glycaemic response variability [11], and better appetite control [12,13]. This phenomenon leads us to hypothesise that lower insulin responses exhibited by these supplements could promote satiety and fullness [10,12,13,14].

Nutritional approaches to type 2 diabetes usually include novel strategies in dietary advice, especially in oral nutritional supplements (ONS) prescription as part of the management of some DM comorbidities or as a complement for daily diet [15,16]. Any ONS designed for people with diabetes (ONS-D) provides better control in postprandial glucose and glycated haemoglobin (HbA_1_c) when compared with standard supplements [17], since they are lower in total carbohydrate content (with a variety of sugar substitutes) and enriched with fibre and monounsaturated fatty acids [16,18,19]. Dietary fat and carbohydrate modifications modulate postprandial glycaemic responses by a reduction in glucose absorption rate [20].

The increase in peripheral glucose uptake via entero-insular axis peptides (EIAPs) such as the glucose-dependent insulinotropic peptide (GIP), glucagon-like peptide 1 (GLP-1), and insulin are a group of synergistic pathways counteracting undesirable glucose postprandial peaks [21]. Furthermore, a decrease in GIP secretion drives to adipocyte hypertrophy arrest and insulin resistance amelioration [22]. By contrast, GLP-1 has a direct suppression effect on appetite and protects pancreatic β-cells from programmed cell death [22,23]. It is well known that beverages have a faster gastric emptying and intestinal transit speed than solid food, which results in a rapid glycaemic response and lower perception of satiety [13,23]. For these reasons, the strategies of specific oral supplements designed for people with diabetes should include adaptation in the overall nutrients content [18,19]. Therefore, variations in both EIAP action and gastric emptying modulation by diet could play a fundamental role in short-time appetite regulation and energy intake [24,25].

Use of ONS-Ds in malnourished or sarcopenic diabetic patients enhances energy intake and overall nutritional status, improving glycaemic control, and thus, cause indirect economic benefits [19]. A meta-analysis by Elia et al. [15] on a total of 23 studies and 784 patients receiving oral supplements or tube feeding showed that when compared with standard supplements, ONS-D significantly reduced postprandial rise in blood glucose, peak blood glucose concentration and glucose area under the curve (AUCG) with no significant effects on HDL, total cholesterol, or triglyceride levels. Furthermore, this study reported a reduced insulin requirement (26–71% lower) and fewer complications in patients with ONS-D therapy when compared with standard nutritional supplements [15]. Therefore, in order to tighten glycaemic control, starch modification and sugar substitution [16,26] has become a primary strategy in the formulation of these supplements [15,16]. However, there is a compelling need to conduct more studies in special situations such as hospitalised patients, older people with DM2 or end-stage kidney disease and patients with cancer [27,28].

Recent evidence suggests a favourable glycaemic response after nutritional supplement intake with sucromalt (a natural analogue of sucrose with a lower glycaemic response) in diabetic patients [18,29,30]. Isomaltulose, another sucrose replacer, is a disaccharide composed of glucose and fructose linked by an alpha-1,6-glycosidic bond exhibiting prolonged absorption, LGI (GI = 32), and a 20–25% lower hydrolyzation rate when compared with sucrose [31]. Interestingly, GIP and GLP-1 secretion are affected by this disaccharide [32], resulting in a better insulin secretion profile [33] and a possible reduction in postprandial appetite [34].

Moreover, a study by Pfeiffer et al. [32] gave evidence on the relationship between high-glycaemic index carbohydrates and a faster GIP release pattern in patients with fatty liver disease, subclinical inflammation, DM2 and cardiovascular diseases. On the other hand, LGI carbohydrate consumption would induce a lower GIP release and a higher release of GLP-1 [32], promoting a better metabolic markers profile in both healthy and type DM2 individuals [18]. Therefore, these authors propose the GIP release rate as a determining factor in the “metabolic quality” and in consequence, relevant criteria for the selection of dietary carbohydrates [32].

Several studies in DM-2 subjects have explored incretin release after consumption of oral nutritional supplements with sucromalt or isomaltulose [18,35]; however, GI and GL have only been studied in healthy subjects [35,36] and not in diabetic patients. Likewise, the correlation between the glycaemic response (GI/GL) and SA as well as EIAP behaviour is not sufficiently well described to date, especially during ONS-D intake, digestion and absorption time.

Based on available literature, we hypothesised that an ONS-D that contains slow-digesting carbohydrates (isomaltulose or sucromalt) resulting in a significant release of GLP-1 and lower secretion of both GIP and insulin. As consequence, a reduction in GI/GL index and subjective postprandial appetite ratings would be found when compared with standard nutritional supplements. Thus, the aim of this study was to assess sucromalt/isomaltulose ONS-D effects on the glycaemic response (GI/GL), EIAP release and postprandial SA in type 2 diabetic individuals.

## 2. Materials and Methods

### 2.1. Study Design

#### 2.1.1. Design and Ethics Issues

A randomised, double-blind, cross-over study was conducted according to Good Clinical Practice Guidelines, applicable Food and Drug privacy regulations and ethical principles based on the World Medical Association-Helsinki Declaration [37]. This research was approved by the Human Research Ethics Committee of the Endocrine and Metabolic Diseases Research Centre (EMDRC), “Dr. Félix Gómez”, School of Medicine at the University of Zulia, Venezuela, and then registered in Clinical Trials.gov (https://clinicaltrials.gov/ct2/show/NCT03829800).

#### 2.1.2. Inclusion and Exclusion Criteria

This study included both male and female DM2 subjects over 50 years old who attended the outpatient diabetes medical clinic at EMDRC [37]. The only antidiabetic therapies allowed were diet/physical activity and/or metformin monotherapy. Body mass index (BMI) between 18.5 kg/m^2^–35 kg/m^2^ was the only compelling anthropometric marker in order to be included in this trial. Patients with diabetes mellitus type 1 (DM1), diabetic ketoacidosis, hypothyroidism/thyrotoxicosis congestive heart failure, gastric, kidney or hepatic diseases, myocardial infarction, stroke and subjects with insulin therapy or sulfonylureas, antibiotic therapy or corticosteroids, end-stage organ failure, or individuals with organ transplantation, coagulation disorder, bleeding disorders, chronic infectious disease (such as tuberculosis, hepatitis B or C or HIV) were excluded.

#### 2.1.3. Population, Sample Size, and Patient’s Selection

Taking into account the criteria mentioned above, the whole EMDRC electronic medical record database was filtered obtaining a population of 57 eligible patients. Literature regarding GI and GL suggests 8 to 10 subjects for a proper meal/supplement assessment [10,36,38]. Thus, a random selection of 23 DM2 patients was made with the purpose of obtaining a sample size with a reasonable accuracy in determining the ONS-D glycaemic impact, GI and GL [36,38]. Since postprandial glycaemia and glycaemic index was our primary outcome, this study was powered to detect the difference among the AUCGs after consuming three oral nutritional supplements (effect size = 0.79) [20,24,35,36,38]. Based on our calculation, at least 15 participants were needed to detect this effect size at 80% statistical power using a cross-over study design [24]. Assuming an attrition rate of 30%, 23 participants were recruited in this study. Eligible subjects were contacted by phone and invited to attend a medical screening visit in order to: (1) be invited to participate in the study, (2) verify if the participant met the inclusion criteria and, (3) asked to give their written consent before beginning the study.

#### 2.1.4. Anthropometric Assessment

Anthropometric data were obtained in fasting state, using light clothing and no shoes. For weight determination and electric bioimpedance study, an UM-018 Digital Scale (Tanita, Tokyo, Japan) was used. Height was measured using a SECA 216 stadiometer (Hamburg, Germany). Body mass index was calculated using the equation: BMI (kg/m^2^) = mass (Kg)/height (m^2^).

### 2.2. Study Protocol

#### 2.2.1. Oral Nutritional Supplements Composition

In this study, three oral nutritional supplements were examined: (1) non-diabetes-specific standard oral nutritional supplements (ET; Ensure^®^ Abbott Nutrition, Columbus, OH, USA); (2) oral supplements with a blend of slow-digesting carbohydrates including resistant maltodextrin and sucromalt (GS; Glucerna SR^®^ Abbott Nutrition, Columbus, OH, USA); and (3) oral supplements composed of lactose, isomaltulose, and resistant starch (DI; Diasip^®^ Nutricia Advanced, Medical Nutrition, Dublin, Ireland).

The macronutrient composition of these formulas per 100 mL is shown in Table 1. Considering that two of the supplements contained a relatively low amount of total carbohydrates, a standardised portion with 25 g of this nutrient was administered in each patient for all tests. This criterion is recommended when the carbohydrate load in the food is low, in order not to overestimate portion size [35,38]. Therefore, all supplements were compared with a glucose load of 25 g (GB), as a reference food (anhydrous glucose dissolved in 250 mL plain water) (100 Kcal) [35,38]. It is important to mention that there was no significant difference in the volume of formulations supplied in this study (Table 2).

The three oral nutritional supplements delivered energy ranging from 149 to 223 kcal (Table 2). Both DI and GS contained 208 kcal per 200 mL versus 205 Kcal per 220 mL, respectively (recommended daily serving size). Supplement DI had a lower percentage of carbohydrate (47 energy%) and protein (19%) but higher percentage of fat (32 energy%, of which 18.5% was monounsaturated fatty acids (MUFAs). On the other hand, the composition of GS was more comparable with DI; a lower percentage of carbohydrate 47.7% and protein 18.42%, and higher percentage of fat 33.81% (MUFA: 20.5%), while the composition of ET had a higher percentage of carbohydrate: 55.68%; lower percentage of fat: 29.45%; and protein: 14.87%; with 250 Kcal per 237 mL. The GI values were calculated according to the information reported in the nutritional labelling of each supplement.

#### 2.2.2. Experimental Protocol

##### Background Diet, Physical Activities and Other Measurements

Subjects were informed about diet and physical activity restrictions to be followed before each session, which included: (1) 10–12 h fasting, (2) abstinence from alcohol, caffeine or smoking and not exercising excessively 24 h before each session; (3) avoiding the metformin morning dose or other medications allowed on trial days until instructed, and to do so at the health centre. Participants were evaluated before each treatment by a licenced nutritionist. In this evaluation, patients had to submit a 3 day food record in order to confirm adherence to the meal plan. The day before the administration of supplements, the nutritionist recommended a standardised dinner before 21:00 and were asked not to consume anything before arriving at the laboratory except water, which was allowed until midnight [14,39]. In order to ensure that participants complied with established protocols, they had to complete a compliance survey. In case they did not comply with the previous test protocols, the test sessions were rescheduled.

During the appetite test sessions, patients engaged in 60 min of sedentary activities (word puzzles, reading, board games, etc.) [14,39]. The activities were performed in a friendly, non-competitive manner to avoid emotional excitement or stress. Any food-related topics were avoided for the duration of the sessions [14]. Research team members evaluated the compliance of the experimental protocol verifying the correct administration to all patients in each visit. Participants had access to water during the day of the trial. The leading investigator reviewed these records before performing the food tolerance test. During each test day subjects were allowed to drink water each hour (maximum 150 mL each hour) immediately after filling in the appetite questionnaires. Water consumed during the test session on the first day was recorded and repeated on the other test days [14,39].

##### Randomisation

All participants were randomly assigned to eight consumption tests: two for the standard glucose solutions and two for each of the three nutritional supplements. This scheme was carried out with an interval of 1 week between tests in random sequences. The test supplement selection was randomised using a computerized randomisation matrix. The order of supplement was further randomised for each subject. The number of tests in each patient was done according to methodological considerations for glycaemic index protocols [38]. Appetite was assessed twice for the same subject before and after supplement intake on two different occasions [14,39].

Previous-evening lunch standardisation: the dinner consumed by all participants the night before each session day consisted of meat with boiled rice and a fruit cake for dessert ∼505 kcal [14]. The energy content of this meal was 35% of the daily estimated energy needs of each participant [40]. The distribution of energy in the evening meal was 50% from carbohydrate, 37% from fat and 13% from protein [14,39].

### 2.3. Measurement of EIAP/GI and Subjective Appetite Evaluation

#### 2.3.1. EIAP and Glycaemic Index

Participants attended the EMDRC following 10–12 h fasting at 07:00. Both duplicated blood capillary samples (0.5 mL) and venous samples were taken in basal state, and then each patient was randomly assigned to drink one of both the ONS (ET DI or GS) supplement or the reference food (glucose 25 g in plain water) during a period not exceeding 15 min [35,36,38]. The reference food (GB) (glucose solution) was used for glucose and insulin AUC determination only. Subsequently, samples of capillary and venous blood were obtained at 30, 60, 90, 120 and 180 min for serum glucose, insulin, GIP and GLP-1 measurement. During this phase, subjects were comfortably seated in a room with a quiet environment [38]. This process was repeated seven more times, on different days, with one week interval until all consumption tests were done [36,38].

#### 2.3.2. Subjective Appetite Assessment

The appetite sensations measured in this study were: hunger, desire to eat, prospective food consumption and fullness assessed on different days from those in which the GI was determined. The visual analogue scale (VAS) chart was supplied in every session [39] and the subjects were asked to fill this instrument at baseline (0 min), 30, 60, 90, 120, 150 and 180 min after the ingestion of each supplement. This instrument contemplates four questions: What is your feeling of fullness? How hungry are you? How intense is your desire to eat? And how much food do you think you could eat? [14,39].

The VAS structure consisted of 100 mm lines anchored at each end with opposite statements with a scale of 0 to 100 mm, in which 0 means absence of perception and 100 maximum perception. The distance between 0 and the marked point (an “x” placed by the participants on the line to indicate their assessment at that time) was measured to quantify the perceived sensations. The score was calculated by measuring the distance in millimetres from the beginning of the line to the “x” position (from left to right) [39].

The following equation was used to calculate the ratings of subjective appetite [41]: “Subjective appetite = (desire to eat + hunger + (100 − fullness) + prospective food consumption)/4”.

#### 2.3.3. Laboratory Determinations

Capillary glycaemia was determined by the glucose oxidase method using a portable glucometer (Optium Xceed, Abbott Laboratories, Dallas, TX, USA) Both intra-assay and inter-assay coefficients of variation were 3.2 and 10.8%, respectively. Plasma Insulin (mU/L) was measured by an enzyme-linked immunosorbent assay (10-1113; Mercodia, Uppsala, Sweden) with a minimum detectable limit of 1.0 mU/L, and an intra- and inter-assay variation coefficients of 3.0% and 8.7%. Glycated haemoglobin HbA1c was determined using a cationic exchange resin separation method (SIGMA, St. Louis, MO, USA). Plasma total GIP (pg/mL) and GLP-1 (pmol/L) were measured by radioimmunoassay (RIA) (SIGMA, St. Louis, MO, USA). The minimum detectable limits were 2 pmol/L and 3 pmol/L with an intra- and inter-assay coefficient of variation for GIP of 3.9% and 9%, and for GLP-1 6.3% and 10.3%, respectively. Total cholesterol, triacylglycerides, and HDL-C were determined by commercial enzymatic-colorimetric kits (Human Gesellschaft für Biochemica und Diagnostica MBH, Wiesbaden, HE, Germany). Serum LDL-C levels were calculated according to Friedewald’s equation.

### 2.4. Data Processing and Statistical Analyses

Statistical analyses were performed using IBM SPSS Statistics for Windows, version 23.0 (IBM Corp., 2015, Armonk, NY, USA). Shapiro–Wilk test was used for the normality distribution assessment of quantitative variables.

Incremental areas under the curves (IAUCs) were determined according to the trapezoidal method for all variables using NCSS statistical software version 12.0 (NCSS, LLC, 2018, Kaysville, UT, USA). A 2 h glycaemic response curve was generated for each subject for test foods. Any area below the baseline fasting value was ignored. The calculated median of AUCG for three test foods from 16 participants was compared with the response to reference food or glucose solution (median of two measurements), and the GI value of the glucose solution was set as 100.

The GI was calculated using the following equation [36,38]: GI = (AUCG value for the test food/AUCG value for the reference food) × 100.

Data obtained was classified in low GI (≤55), intermediate (55–69) and high (≥70) [41]. Glycaemic load (GL) was represented by a derivative measure of the GI of the nutritional supplement tested and calculated by the following formula: GL = (GI × grams of carbohydrate per food portion)/100 [36,38].

All quantitative variables were presented as mean ± standard error of the mean (SEM). Plasma glucose, insulin, GIP, GLP-1, perceptions of hunger, desire to eat, prospective food consumption, fullness and SA had a normal distribution and its arithmetic means were analysed using ANOVA for repeated measures with the Tukey’s HSD (honestly significant difference) test. Significant statistical differences between ONS were evaluated through one-way ANOVA. The bivariate relation between variables such as the AUC, blood glucose, EIAP and SA was analysed by correlation coefficients for each oral test. Statistical significance was accepted at *p* < 0.05.

## 3. Results

At the beginning of the study, the initial sample was 23 individuals (12 women, 11 men), but seven subjects did not complete the trial for different reasons: (1) two subjects needed both corticosteroid and antibiotic therapy; (2) two subjects initiated a vigorous physical activity program by medical prescription; and (3) three voluntarily withdrew from the study. At the end of the study, only 16 subjects (seven women and nine men) completed all the test protocols. Table 3 shows the general characteristics of the sample.

The protocol was well tolerated by all subjects. No individual reported nausea, dizziness or vomiting after taking the nutritional supplements or the reference product. Basal concentrations of serum glucose, insulin, GLP-1 and GIP (Table 4 and Figure 1) did not show significant differences according to sex or among weekly visits. Similarly, hunger perception, fullness, desire to eat, prospective food consumption in fasting state and SA ratings (Table 5 and Figure 2) did not show significant differences among gender or study session (*p* > 0.05).

### 3.1. Glycaemic Response and EIAP Concentrations

Glycaemic curves, as well as the mean and SEM of the glucose AUC_0–180 min_ after ingestion of both the reference product (glucose) and the nutritional supplements, are shown in Figure 1. Glucose maximum peak was observed at 60 min (Table 4) for all products but significantly higher for GB 15.03 ± 0.20 when compared with ET 10.80 ± 0.12mmol/L (*p* < 0.001), DI 10.17 ± 0.05 (*p* < 0.001) and GS 9.10 ± 0.06 mmol/L (*p* < 0.001). Glucose at 180 min was significantly higher in comparison with both, fasting level for GB (*p* < 0.001) and for ET (*p* < 0.024). The AUC_0–180 min_ of ET (*p* < 0.001) was significantly higher than DI (*p* < 0.001) and GS (*p* < 0.01) (Figure 1).

#### 3.1.1. Insulin

Plasma insulin concentrations increased after the consumption of all supplements and the reference product, reaching significant differences at 90 min for GB (serum peak) in comparison to ET (*p* < 0.05), GS (*p* < 0.001) and DI (*p* < 0.001), respectively (Figure 1). At 150 min, ET presented a higher glucose concentration than DSF (*p* < 0.001), but no significant differences were found in insulin concentration between DI and GS (*p* = 0.976), see Table 4. The AUC_0–180 min_ in insulin response was significantly lower in GS when compared with the other supplements (*p* < 0.001) (Figure 1).

#### 3.1.2. GLP-1

Maximum GLP-1 concentration was observed at 30 min after the intake of the three supplements, significantly higher for GS in comparison to ET (*p* < 0.05) and DI (*p* < 0.05). At 150 min, concentrations of GLP-1 in ET and DI supplements were similar (*p* = 0.841), but the value of this incretin was significantly higher for GS when compared with both, ET (*p* < 0.001) and DI (*p* < 0.001), (Table 4). The AUC_0−180 min_ of the GLP-1 response was significantly higher in GS in contrast to the ET (*p* < 0.001) and DI (*p* < 0.001), (Figure 1).

#### 3.1.3. GIP

The GIP plasma concentration increased after consumption of all supplements. The maximum peak of this incretin was observed at 90 min with ET and DI, which was higher when compared to GS levels (*p* < 0.05). At 150 min, ET presented higher GIP concentrations when compared to GS (*p* < 0.001) and DI (*p* < 0.001), however, DSF levels did not show significant differences (*p* = 0.844), Table 4. The AUC_0–180 min_ of the GIP response for GS was lower when compared to DI (*p* < 0.001) and ET (*p* < 0.001, (Figure 1).

### 3.2. Subjective Appetite Measurements

Hunger sensation, fullness, desire to eat, prospective food consumption and SA from baseline to 180 min are shown in Figure 2. Consumption of the different treatments promoted an immediate decrease in hunger and desire to eat accompanied by an increase in the perception of fullness, reversing these sensations over the curve as time passed.

The arithmetic mean of hunger perception decreased after the consumption of all supplements, registering the lowest level at 30 min for E, while the minimum value for GS was evidenced at 90 min, significantly lower when compared to ET (*p* < 0.05) and DI (*p* < 0.05), (Table 5). The AUC_0–180 min_ of hunger sensation for GS was significantly lower when compared to ET (*p* < 0.001) and DI (*p* < 0.001), (Figure 2).

Regarding fullness sensation, the maximum level was found at 30 min in the three groups, without significant differences between DI and GS, while the peak of fullness sensation was significantly lower with ET (*p* < 0.05), (Table 5). The AUC_0–180 min_ of this sensation was significantly higher in GS when compared to DI (*p* < 0.001) and ET (*p* < 0.05), (Figure 2). On the other hand, the desire to eat AUC_0–180 min_ was significantly lower for GS when compared to DI (*p* = 0.035) and ET (*p* < 0.001), (Figure 2).

This same pattern was evidenced in the prospective food consumption, in which the AUC_0–180 min_ was significantly lower in GS when compared with ET (*p* < 0.001) and DI (*p* < 0.001), (Figure 2). Subjective appetite SA decreased to a minimum value at 30 min and then increased 60 min after the three treatments for all subjects; this score was higher with ET when compared to GS (*p* < 0.01) and DI (*p* < 0.01), (Table 5). The AUC_0–180 min_ of SA was significantly lower with GS than with DI (*p* < 0.01) and ET (*p* < 0.001), (Figure 2).

### 3.3. Correlation Analysis Between EIAP, Serum Glucose and Subjective Appetite

After ET intake, insulin concentration AUC_0–180 min_ and subjective sensation of fullness were directly related (*r* = 0.713, *p* = 0.021), while an inverse relationship between fullness perception and GLP-1 concentration AUC_0–180 min_ (*r* = −0.756, *p* = 0.011) and serum glucose (*r*= − 0.687; *p* = 0.028) was observed. The value SA was directly correlated with serum glucose (*r* = 0.659, *p* = 0.038), see Table 6. No statistically significant correlations were found for AUC_0–180 min_ concentrations of these peptides with DI and GS.

Correlations between baseline and postprandial concentrations at 30, 90 and 120 min of glucose, EIAP and SA measures were accomplished in all treatments. Insulin at 30 min for ET was inversely related to hunger sensation (*r* = −0.745, *p* = 0.012) and SA (*r* = −0.849, *p* = 0.002) (Appendix A). DI intake was correlated with glycaemia and prospective food consumption at 30 min (*r* = 0.775, *p* = 0.008) and, GLP-1 with the desire to eat at 120 min (*r* = 0.667 *p* = 0.035); whereas, SA was inversely correlated at 30 min with GIP (*r* = −0.688, *p* = 0.028) (Appendix A).

GS evidenced a direct relationship between glycaemia at 90 min with sensation of fullness (*r* = 0.698, *p* = 0.025) and levels of GIP with sensation of hunger (*r* = 0.825, *p* = 0.003). GLP-1 at 30 min and prospective food consumption were inversely related (*r* = −0.722, *p* = 0.018). SA was directly correlated with blood glucose levels at 30 min (*r* = 0.711, *p* = 0.021) (Appendix A).

### 3.4. Glycaemic Index and Glycaemic Load

ET presented a GI mean higher than that calculated for DI (*p* < 0.001) and GS (*p* < 0.001), respectively. Comparing both specific supplements for diabetics, the lowest value for this indicator was evidenced in GS (*p* < 0.001). Concerning GL, ET showed the highest mean compared to the rest of treatments (*p* < 0.001), and the lowest mean value for DI (11.28 ± 0.14, *p* < 0.001), (Table 7).

### 3.5. EIAP and SA Relation with GI and GL

In relation to each supplement, it was found that hunger sensation AUC_0–180 min_ was directly correlated with GI (*r* = 0.777, *p* = 0.008) and GL (*r* = 0.777, *p* = 0.008) for ET; while DI, both GI and GL were inversely related to GIP (*r* = −0.867, *p* = 0.001). GS, GI and GL were inversely related with fullness sensation (*r* = −0.698, *p* = 0.025). SA ratings did not correlate significantly with any of these indexes (*p* > 0.05), (Appendix A).

## 4. Discussion

This study assessed ONS-D with isomaltulose and sucromalt versus a standard oral supplement on GI/GL, insulin response, incretin release and SA in DM2 patients. In this regard, the main finding of this study confirmed that ONS-D intake in diabetic subjects stimulated GLP-1 release, reducing GIP levels with a subsequent decrease in insulin secretion. This particular EIAP pattern promotes a lower IG/CG index when compared with a standard supplement. In spite of the former, there was a reduction of SA and AUC_0–180 min_ after ONS-D intake; only correlation between hunger perception, fullness and some metabolic variables were found after GS intake.

These findings confirm that ONS-D consumption promotes a better metabolic profile in diabetic subjects than standard supplements, allowing greater control in postprandial appetite. Specifically, this investigation demonstrates that plasma glucose levels and glucose AUC_0–180 min_ were significantly lower after the ingestion of ONS-D than ET. Our observations are consistent with previous research carried out with slow-digesting carbohydrates supplements [18,29,42,43]. In this study, after the consumption of GS, the mean glycaemic peak was consistent with ADA recommendations for glucose level < 180 mg/dL (9.99 mmol/L), with elevated HbA1c in DM patients [7] and IDF of 160 mg/dL goal, both in the postprandial period [6]. Similar to observations by Mottalib et al. [42] this study shows that serum glucose level after ONS-D ingestion returned to baseline in a shorter period (150 min) when compared with ET (180 min) [42], see Figure 1 and Table 4 [18,29,42]. These differences in glycaemic profile can be attributed, at least in part, to the low GI of ONS-D [18,29,35,42], a point of paramount importance to avoid cardiovascular complications [44] because of pro-inflammatory cytokines and oxygen free radical overproduction [44,45]. In this trial, the consumption of GS produced lower values of GI/GL, a lower increment of GIP/insulin and more significant release of GLP-1.

Among the different factors that influence the GI of a food, the source and carbohydrate type are very relevant aspects. High GI carbohydrates differ from those with LGI, not only in postprandial glycaemic and insulinemic response but also in GIP release [45,46,47]. In this regard, Pfeiffer et al. [32], suggest a novel concept that encompasses the intake of LGI CHO with a lower release of GIP and a greater GLP-1 secretion results in improvements of metabolic markers in healthy [45], type 2 and insulin-resistant individuals [46].

This concept relates to GI of each food to different secretory responses of GIP and GLP -1, which are released in different segments of the small intestine [21,45,46,47]. These authors propose that both a fast and pronounced GIP release in the proximal small intestine by high GI carbohydrates programming the intermediary metabolism towards useful energy storage but adversely promoting fatty liver disease [46], insulin resistance [48], obesity [49], subclinical inflammation and hypertriglyceridemia [46]. This program could represent an evolutionary advantage in times that rapid energy storage was required [32,46]. Complete understanding of the pathophysiological mechanisms of foods with a high GI provides a basis for the development of nutritional and therapeutic solutions [10,11,41,46].

Nonetheless, it is essential to differentiate the digestion (di, oligo and polysaccharides) and absorption rate (monosaccharide), from the particular metabolism of each monosaccharide. This is because certain simple sugars, such as fructose, with a relatively low GI (=23) [36], could induce insulin-independent additional metabolic effects [50] on uric acid levels, blood pressure, liver cell triacylglycerides content and hepatic insulin sensitivity when consumed in high amounts [51]. On the other hand, tagatose is a low GI (=3) monosaccharide [36], that promoting a GLP-1 release in a similar extent to fructose without any significant GIP secretion response [52] and exhibiting an interesting glucose-lowering effect [53]. In this study, we assessed two of the most employed slow-digesting carbohydrates in ONS-D: isomaltulose and sucromalt.

Beneficial metabolic effects have been reported when low GI disaccharides = 32, such as isomaltulose [31,36], are added to oral supplements in people with DM2 [29]. This disaccharide has an α-1,6-glycosidic bond replacing the original sucrose´s 1,2-glycosidic linkage by enzymatic isomerisation rearrangement obtained from beet sugar [33]. This molecular reorganisation leads to slower digestion and, in consequence, delayed intestinal uptake of glucose and fructose [33,54]. Unlike sucrose, isomaltulose administration prevents proximal K cells stimulation, secreting less GIP and promoting a smaller insulin release [32,33]. For its part, the low glycaemic response to sucromalt showed a sustained increase in GLP-1 secretion at 4 h post intake, suggesting an almost complete uptake by the small intestine [30]. Thus, it is important to distinguish the effects in GI from those caused by changes in gut microbiota that occur when sugar reaches the colon and alter microbiome composition, affecting long-term carbohydrate metabolism and insulin response [55,56,57].

In this study, ONS-D insulinemic behaviour interestingly showed a lower AUC_0–180 min_ level in ET, especially after GS at 90 min, the time when the maximum peak of this hormone occurs. Meanwhile, the maximum insulin concentration after DI intake occurred at the 60 min (Figure 1). The maximum increase in GIP levels after GS occurred after the rest of the treatments (150 min), but it was only statistically different from ET (Table 4). Likewise, the AUC_0–180 min_ for GIP was lower for ONS-D compared with the ET, and lowest for GS versus DI, (Figure 1). This finding could confirm the theory mentioned above about the effect of slow digestion carbohydrates on the release of insulin and GIP, although insulinemic peak after ET also occurred in 90 min, but with a much higher incretin concentration than that produced by ONS (*p* < 0.05 for both).

It has been proposed that slow-digestion carbohydrates can reach the more distal segments of the small intestine before being absorbed, hence, stimulating a late-plasma increase of GLP-1 [30,52]. In this trial, the AUC_0–180 min_ of GLP-1 was higher after ONS-D consumption when compared to ET, and higher for GS when compared with DI, (Figure 1). Our results were similar to those reported by Devitt et al. [29] regarding metabolic differences after specific supplement ingestion composed of a variety of carbohydrates like tapioca dextrins, isomaltulose, tapioca starch/fructose and sucromalt in DM2 patients. In this study, patients showed an increase in AUC_0–240 min_ for GLP-1 after sucromalt-based supplement intake, but it was only significantly higher after supplements made with tapioca dextrin in comparison with the standard ET [29].

Some benefits of increased GLP-1 secretion in DM2 patients are an improvement in insulin–glucagon ratio, suppression of endogenous glucose production and the increase in first-pass splanchnic glucose uptake [47,58]. It is currently unclear whether inhibition of L-cell secretion or GLP-1 enhanced degradation entails to the characteristic blunted-effect of this incretin in DM2. Also, the exact mechanism of GLP-1 effects on glucose control has not yet been elucidated [18,21]. Although there are studies about this incretin for isomaltulose and sucromalt in healthy subjects [46], other studies have reported benefits in individuals with metabolic syndrome, obesity and DM2 after isomaltulose versus sucrose consumption [31,58,59,60], but few have compared the effects of cross-consumption of pre-loads elaborated with these types of carbohydrates in DM2. To date, only one study has determined a higher release of GLP-1 after the consumption of isomaltulose in individuals with diabetes [58]. Our results showed an AUC_0–180 min_ of GLP-1 higher after the consumption of GS versus DI, (Figure 1) exhibiting a synergistic effect of these carbohydrates.

In this sense, it is well-known that GLP-1 secretion is directly related to macronutrients composition, in particular to both carbohydrates and monounsaturated fatty acids (MUFAs) [61,62] without any significant effect on insulin levels [61,63,64]. This is consistent with our results and with the Mottalib et al. work regarding GLP-1 secretion and MUFA content in ONS-D when compared to ET [42]. In a study by Printz [65], adequate glycaemic and insulinemic responses were found after the intake of three enteral supplements for diabetics in subjects with DM2, but no significant differences in the release of GIP and GLP-1 were found [65]. In fact, carbohydrates used in Printz’s [65] study, such as glucose, fructose, lactose and maltose [65], probably resulted in both changes in the final place of the intraluminal digestion and the speed of absorption, which could explain these results, especially when comparing the forenamed carbohydrates with those administered in our study. Even though DI also contains disaccharides such as lactose, its metabolic profile could be sufficient to produce a more significant GLP-1 release of and less GIP than ET, but not enough to produce a better effect on incretins and insulin than GS. This observation confirms previous findings that both the amount and type of carbohydrates and fats influence incretin release 18,24,29,45], as well as in the GI and GL.

Our study could be one of the first demonstrating ONS-D effects on GI in DM2 subjects. In fact, when compared with glucose solution, the evaluated supplements turned out to have an intermediate GI value in people with diabetes for ET = 56. Meanwhile for GS = 47 and DI = 51, the result was a low GI. Whereas, GL was high for ET = 23 and intermediate for GS = 11 and DI = 12, (Table 7). The mean of these values is higher than the reported in the international GI and GL tables for healthy subjects [36]: GI for ET = 48, GS = 23, DI = 12; with an intermediate GL: ET = 16, GS = 6 DI = 3. In a randomised cross-over study conducted by Hofman et al. [35], in which the GI of 12 supplements was evaluated in healthy subjects, the mean GI value in the ONS-D was 19.4 ± 1.8, and from 42.1 ± 5.9 in standard supplements [35].

Significant differences have been reported in the GI value of different foods and/or typical foods between healthy subjects and DM2 [66]. It is well known that the subject’s characteristics does not have a significant effect on mean IG values [38], but the variation of the values can differ in different groups, being higher in people suffering from type 1 diabetes (29%) than in healthy subjects (22%) or in DM2 patients (15%) [38]. Our results are comparable with a previous study in which the GI for a DM oral specific supplement was assessed in healthy subjects (IG = 27) and DM2 (IG = 54), finding a significant difference between groups [67].

This situation can be explained by a greater relative increase in the glycaemic response after consumption of the reference food (GB) in people with diabetes compared to healthy subjects [38,68]. One possible explanation in that a defective insulin secretion is unable to counteract greater glycaemic excursions in DM2 patients. At the same time [68], healthy people preserve their insulin secretory machinery, preventing a greater glycaemic increase especially for the lowest digestion rates [38,68].

As mentioned above, we found a higher GI for DI, even though a lower GI value has been reported in healthy subjects when compared to the rest of the treatments [35,36]. However, in the study conducted by Hofman et al. [35], both supplements GS and DI contained fructose 1.9 g/100 mL, and a higher amount of MUFAs, DI = 3.6 g/100 mL versus GS = 3.8 g/100 mL than the supplements evaluated in this study, therefore, it is not possible to make an exact comparison [35].

The GI value for DI in DM2 patients could be explained in part by the quantitative sugar content (which has an 8.3 g versus 0.0 g to GS per portion given in this study) (Table 2). The rest of the components like both the amount and type of fibre can also influence these results. Moreover, soluble fibre can decrease the GI by many factors such as postprandial glucose fluctuation cushioning, and, by its action on intestinal motility, on peptide action and gastrointestinal enzymes [29,69,70].

In this study, total fibre concentration in DI was 2 g/100 mL, whose proportion corresponds to the 80:20 ratio of soluble/insoluble fibre compared to GS, whose total amount corresponds to 1.8 g/100 mL soluble (Table 1). In this regard, identical fibre concentrations generated different GI, such is the case of supplements with 1.5 g/100 mL of fibre and with GI = 26 and 17, respectively, in healthy subjects [35,36], constituted by different fibre mixtures based on fructooligosaccharides, inulin, oligofructose, Arabic gum, soybean polysaccharides and cellulose. Furthermore, it has been reported that soluble fibre can stimulate appetite-regulating peptides such as GLP-1 and pancreatic peptide YY (PYY) in rodents as well as in human [69,70,71]. It is important to note that DI has resistant starch, whereas GS contains a modified and resistant maltodextrin linked to soluble fibre [69]. In a study in healthy subjects, an increase in peptide YY concentrations and GLP-1 was observed alongside a corresponding decrease in the sensation of hunger and an increased satiety perception after the consumption of tea with 10 g of this component [69].

Few investigations about OSF-D intake have correlated SA with incretin concentrations as it has been evaluated in this study. This indicator was quantified through a score that included variables such as perception of hunger, desire to eat, prospective consumption of food and fullness [14]. It was observed in this investigation that plasma insulin, GIP and GLP-1 were related to some of these parameters but observing a lower SA ratings AUC_0–180 min_ in the ET (Appendix A). It is known that appetite regulation is a complex process stimulated by several central and peripheral signals in response to energy and, mainly, to food composition, where emotional, sensory and environmental factors can influence the overall response [71,72].

There is a lack of consensus regarding GI/GL usefulness in predicting appetite and food intake [12,73,74]. Although GI is not synonymous with glycaemic response, the debate is anchored to the controversy toward the effect of postprandial glycaemic level and its effects on SA reduction [14,75]. Some authors state that the evidence about these affirmations are not conclusive [73,76], postulating that postprandial glycaemic and appetite are not related and considering that insulin response [74], but not the glycaemic response is the real mediator of the short-term appetite reduction, as shown by Flint et al. [14,76]. Specifically, Flint et al. has reported that the maximum insulinemic peak after meal ingestion was related to a decrease in hunger sensation and a satiety increase.

Likewise, in our study, precisely at 30 min and not at the maximum peak, insulin values were inversely related to the sensation of hunger and the overall of SA rating after ET intake, but not the perception of fullness (Appendix A). However, the same behaviour for these variables was not evident after the consumption of ONS-D. Possibly, this was due to the higher and faster insulin increase produced by ET at this point of the curve, based on the type and amount of non-extended release carbohydrates of this ONS, and more than half corresponded to free sugars (14.1 g/per portion given in this study) (Table 2). Despite this premise, a relationship between glycaemia concentration and prospective food consumption 30 min after DI intake was found, (Appendix A) and a direct relationship between glycaemic levels in 90 min with the sensation of fullness after the consumption of GS (Appendix A). The observed feeling of fullness could be related to another mechanism produced via carbohydrate type and fibre content in GS, a supplement that besides sucromalt also has amylase-resistant maltodextrin, in which fibre-viscosity addition could increase the fullness sensation [69].

It is important to note there was an inverse correlation between GLP-1 levels and prospective consumption of food 30 min after GS ingestion (Appendix A). In other studies [24,77], a relationship between GLP-1 and delayed gastric emptying has been evidenced. This gastrointestinal response would influence the feeling of fullness after GS intake at this curve time. Niwano [12] showed that high GI foods consumption are associated with increased hunger and short-term satiety reduction in humans, but not over the long-term [74].

It is relevant to highlight the inverse correlation between GIP and GI/GL after ONS-D ingestion (Appendix A). This finding could confirm the theory proposed by Pfeiffer et al. [32] regarding the role of this incretin as an indicator of “carbohydrate metabolic quality” [32]. On the other hand, we confirmed our hypothesis that ONS-D with slow-digesting carbohydrates strongly stimulates GLP-1 release with a subsequent decrease in GIP and insulin secretion, promoting a lower IG/CG index in DM2 subjects when compared with a standard supplement. Although ONS-D reduced the AUC_0–180 min_ of subjective appetite, only GS exhibited both a hunger sensation decreasing effect and an increased fullness perception in some points of the postprandial response. Finally, these results were also consistent with Peters et al. [78], who evaluated the digestibility of three carbohydrates on appetite and its relation to blood glucose levels and postprandial insulin, reporting that glycaemic response had minimal effects on appetite when the products only differed in the rate and extension of carbohydrate digestibility [78].

The limitations of our research comprised the lack of evaluation of some variables such as gastric emptying. Although the number of subjects who completed this study was sufficient to assess GI/GL accurately, a higher number of patients is recommended for SA evaluation. On the other hand, one of the strengths of this study is that the results of these indicators, especially GI/GL, could be the first of their kind in the literature done in diabetic individuals from Latin America. It is also one of the first to combine these variables with subjective appetite and incretin levels. It would be a matter of interest to extend the curve’s time after consumption in order to evaluate the intake suppression force to the following meal, along with intervention protocols regarding intestinal microbiota in this type of individual.

## 5. Conclusions

The results of this study showed lower values in postprandial subjective appetite ratings and better metabolic profile after ONS-D intake when compared to standard supplements. A more attenuated glycaemic and insulinemic response along with a lower GIP release and higher levels of GLP-1 confirmed the synergistic effect of slow-digesting carbohydrates along MUFA addition. Isomaltulose and sucromalt may have influenced these factors. In this study, GI/GL in subjects with DM2 after ONS-D consumption were lower than the reference food (glucose solution) and the standard supplement, and lower for GS than DI.

Strategies in food technology, such as intestinal amylase-resistant dextrins along with new functional fibres, need to be considered in low GI product development in order to obtain adequately managed metabolic responses of fullness perception after ONS-D consumption. Our study qualifies two of these supplements as optimal for prescription in people with diabetes when compared with a standard supplement. However, it is necessary to conduct more investigations allowing to correlate long-term appetite suppression with the EG/CG of these supplements.

## Figures and Tables

**Figure 1 nutrients-11-01477-f001:**
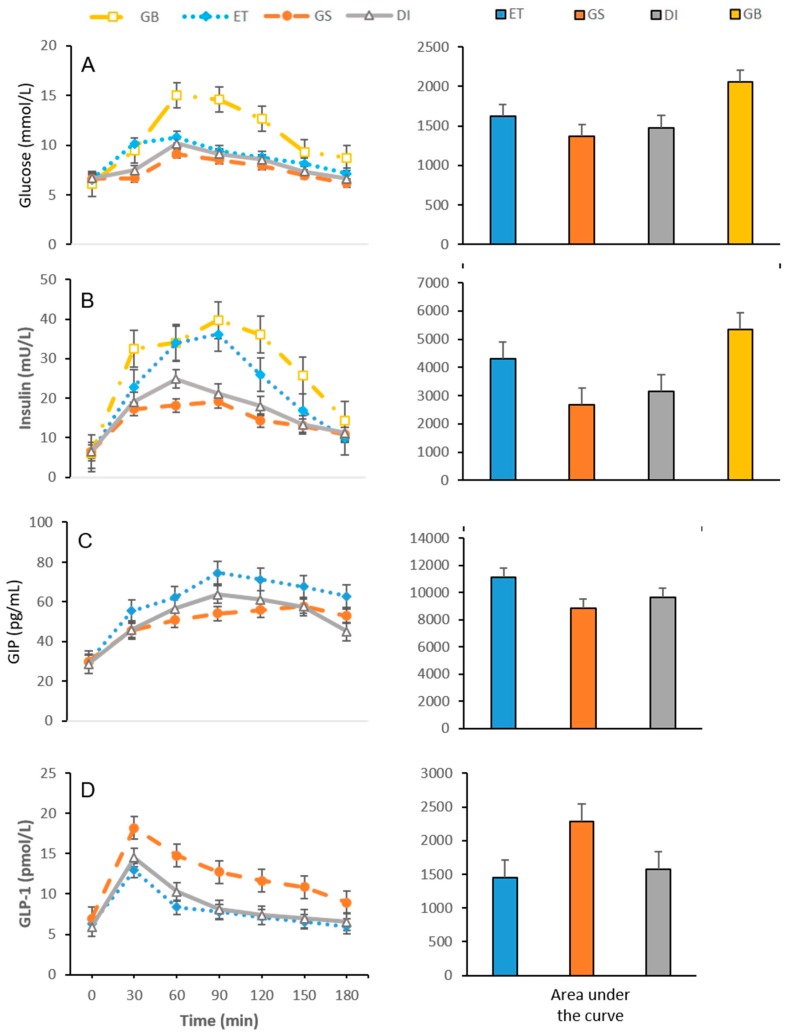
Time course and AUC_0–180 min_ of serum glucose, insulin GIP and GPL-1 concentrations following ingestions of GB, ET, GS and DI. (**A**) Glucose in relation with time and AUC_0–180 min_, (**B**) insulin in relation with time and AUC_0–180 min_, (**C**) GIP in relation with time and AUC_0–180 min_, and (**D**) GLP-1 in relation with time and AUC_0–180 min_ for all the different types of treatments. Data are expressed as means ± SEM; *n* = 16. The same colour scheme was used for all the graphs. All AUC_0–180 min_ means significant differences (*p* < 0.02) in each group. Treatment groups: (GB) Glucose solution or reference product; (ET) standard nutritional supplement not specific for diabetics; (GS) resistant maltodextrin and sucromalt supplement; (DI) isomaltulose and resistant starch supplement. GIP: glucose-dependent insulinotropic polypeptide; GLP-1: glucagon-like peptide 1.

**Figure 2 nutrients-11-01477-f002:**
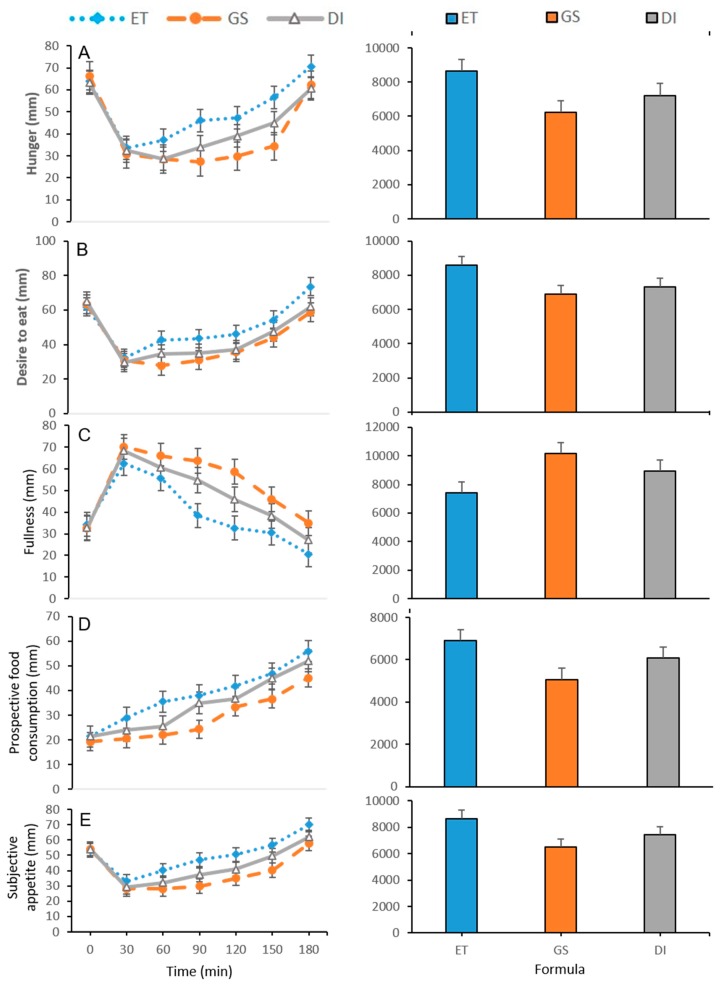
Time course and formula of postprandial perception of hunger, desire to eat, fullness, prospective food consumption, subjective appetite and AUC_0–180 min_ values following ingestions of GB, ET, GS and DI. (**A**) hunger in relation with time and formula; (**B**) desire to eat in relation with time and formula, (**C**) fullness in relation to time and formula, (**D**) prospective food consumption in relation with time and formula and (**E**) subjective appetite in relation with time and formula. Data are expressed as means ± SEM, (*n* = 16). Data comparisons about differences in subjective measurements of appetite according to consumption tests are described in Table 5. The same colour scheme was used for all graphs. All means of AUC_0–180 min_ showed significant differences (*p* < 0.02) in each group. Treatment groups: (ET) standard nutritional supplement not specifically for people with diabetes. (GS) Resistant maltodextrin and sucromalt supplement. (DI) Isomaltulose and resistant starch supplement.

**Table 1 nutrients-11-01477-t001:** Macronutrient composition of the oral nutritional supplements per 100 mL.

Composition	ET	DI	GS
Calories (kcal)	105	104	93
Protein (g)	3.8	4.9	4.3
Fat (g)	2.5	3.8	3.5
Saturates (g)	0.4	0.5	0.3
Monounsaturates (g)	0.8	2.2	2.1
Polyunsaturates (g)	1.3	1.1	0.9
Total carbohydrate (g)	17.3	11.7	10.9
Sugar (g)	10.0	8.3	1.7
Dietary Fibre (g)	1.0	2.0	1.8
Soluble (g)	0.0	1.6	1.8
Non-soluble (g)	0.0	0.4	0.0
Chromium (µg)	12.7	12.0	5.0
Portion size (mL)	100	100	100

Ensure^®^ (ET): A standard oral nutritional supplement which is non-specific for diabetic patients. Diasip^®^ (DI): Isomaltulose and resistant starch oral nutritional supplement. Glucerna^®^ (GS): Resistant maltodextrin and sucromalt oral nutritional supplement.

**Table 2 nutrients-11-01477-t002:** Nutrient composition of oral nutritional supplements based on 25 g available carbohydrate.

Composition	ET	DI	GS
Calories (kcal)	149	223	214
Protein (g)	5.3	10.5	9.9
Fat (g)	3.5	8.1	8.1
Saturates (g)	0.5	1.0	0.6
Monounsaturated (g)	1.1	4.7	4.9
Polyunsaturated (g)	1.8	2.3	2.0
Total carbohydrate (g)	25.0	25.0	25.0
Sugar (g)	14.1	8.3	0.0
Dietary Fibre (g)	0.9	4.3	4.1
Soluble (g)	0.0	3.4	4.1
Non-soluble (g)	0.0	0.8	0.0
Chromium (µg)	12.4	25.8	11.5
Portion size (mL)	141	214	230

Ensure^®^ (ET): A standard oral nutritional supplement not specific for diabetic patients. Diasip^®^ (DI): Isomaltulose and resistant starch oral nutritional supplement. Glucerna^®^ (GS): Resistant maltodextrin and sucromalt oral nutritional supplement.

**Table 3 nutrients-11-01477-t003:** General characteristics of the enrolled patients.

	Sex					
	Female		Male		Total	
	Mean	SEM *	Mean	SEM	Mean	SEM
Age (years)	54.75	1. 65	57.83	1.35	56.0	1.11
Weight (cm)	87.75	3.73	90.17	1.22	89.0	1.58
Height (m)	1.68	0.04	1.69	0.01	1.68	0.01
BMC (kg/m^2^)	30.90	0.44	31.04	0.36	30.8	0.26
Waist circumference (cm)	106.00	1.58	111.00	0.89	106	0.77
Base glycaemia (mmol/L)	7.51	0.40	6.75	0.30	7.05	0.26
Total cholesterol (mg/dL)	209.60	5.71	213.77	8.84	212.10	5.56
High-density lipoprotein (mg/dL)	44.70	4.63	44.30	2.32	44.46	2.16
Low-density lipoprotein (mg/dL)	130.95	4.37	133.28	1.54	132.35	1.87
Triglycerides (mg/dL)	161.70	2.56	158.06	4.45	159.52	2.80
Glycated haemoglobin HbA1c (%)	6.95	0.30	6.98	0.30	6.97	0.20

* Standard error of the mean. There were no significant differences between sexes.

**Table 4 nutrients-11-01477-t004:** Serum glucose concentration and EIAP according to each treatment.

Supplement	Time (min)	Serum Glucose (mmol/L)	Insulin (mU/L)	GLP-1 (pmol/L)	GIP (pg/mL)
	0	6.52 ± 0.07	6.44 ± 0.32	6.26 ± 0.28	29.64 ± 0.50
	30	10.14 ± 0.07 ^bc^	22.84 ± 1.00 ^bc^	12.93 ± 0.21 ^bc^	55.44 ± 0.58 ^bc^
	60	10.80 ± 0.12 ^bd^	33.91 ± 0.97 ^bc^	8.35 ± 0.22 ^bc^	62.27 ± 0.89 ^bc^
**ET**	90	9.39 ± 0.16 ^b^	36.20 ± 0.64 ^bc^	7.78 ± 0.15 ^b^	74.68 ± 0.72 ^bc^
	120	8.76 ± 0.17 ^b^	25.90 ± 0.70 ^bc^	7.16 ± 0.27 ^b^	71.23 ± 0.36 ^bc^
	150	8.14 ± 0.21 ^bc^	16.80 ± 0.56 ^bc^	6.55 ± 0.12 ^b^	67.77 ± 0.50 ^bc^
	180	7.13 ± 0.21 ^b^	9.91 ± 0.81	5.99 ± 0.21 ^b^	62.86 ± 1.26 ^bc^
	0	6.64 ± 0.08	6.49 ± 0.11	6.95 ± 0.36 ^c^	29.97 ± 0.40
	30	6.68 ± 0.13 ^ac^	17.24 ± 0.31 ^a^	18.16 ± 0.26 ^ac^	45.57 ± 0.42 ^a^
	60	9.10 ± 0.06 ^ac^	18.16 ± 0.24 ^ac^	14.75 ± 0.24 ^ac^	50.85 ± 0.15 ^ac^
**GS**	90	8.53 ± 0.07 ^ac^	19.09 ± 0.20 ^a^	12.73 ± 0.24 ^ac^	54.23 ± 0.21 ^ac^
	120	7.92 ± 0.05 ^ac^	14.38 ± 0.16 ^ac^	11.63 ± 0.14 ^ac^	55.99 ± 1.09 ^ac^
	150	6.98 ± 0.12 ^a^	12.93 ± 0.19 ^a^	10.86 ± 0.19 ^ac^	57.87 ± 0.31 ^a^
	180	6.16±0.10 ^a^	10.87 ± 0.18	8.91 ± 0.21 ^ac^	52.85 ± 1.69 ^ac^
	0	6.66 ± 0.10	6.50 ± 0.40	6.20 ± 0.53	28.70 ± 1.07
	30	7.47 ± 0.12 ^ab^	19.04 ± 0.27 ^a^	14.51 ± 0.22 ^ab^	46.00 ± 0.71 ^a^
	60	10.17 ± 0.05 ^ab^	24.86 ± 0.35 ^ab^	10.26 ± 0.11 ^ab^	56.51 ± 1.12 ^ab^
**DI**	90	9.10 ± 0.05 ^b^	21.14 ± 0.36 ^a^	8.09 ± 0.17 ^b^	63.78 ± 0.63 ^ab^
	120	8.56 ± 0.07 ^b^	17.98 ± 0.30 ^ab^	7.35 ± 0.14 ^b^	61.10 ± 0.51 ^ab^
	150	7.33 ± 0.07 ^a^	13.24 ± 0.25 ^a^	6.95 ± 0.09 ^b^	57.52 ± 0.50 ^a^
	180	6.61 ± 0.12	11.13 ± 0.21	6.56 ± 0.20 ^b^	45.19 ± 0.96 ^ab^

Treatment groups were defined as a standard nutritional supplement not specific for people with diabetes (ET); resistant maltodextrin and sucromalt supplement (GS); isomaltulose and resistant starch supplement (DI). GIP: glucose-dependent insulinotropic polypeptide; GLP-1: glucagon-like peptide 1; SEM: standard error of the mean. ^a^
*p* < 0.05 versus ET. ^b^
*p* < 0.05 versus GS. ^c^
*p <* 0.05 versus DI.

**Table 5 nutrients-11-01477-t005:** Subjective appetite measurements according to each treatment.

Supplement	Time (min)	Hunger (mm)	Fullness (mm)	Desire to Eat (mm)	Prospective Food Consumption (mm)	SA (mm)
	0	63.80 ± 1.70	34.30 ± 0.88	61.80 ± 1.14	21.40 ± 1.88	53.18 ± 0.40
	30	33.60 ± 2.45	62.50 ± 1.60 ^b^	32.10 ± 1.19	28.90 ± 2.04 ^b^	33.03 ± 0.91 ^bc^
	60	37.10 ± 1.10 ^bc^	55.70 ± 2.32 ^b^	42.60 ± 1.11 ^bc^	35.50 ± 0.91 ^bc^	40.08 ± 0.56 ^bc^
**ET**	90	46.00 ± 1.00 ^bc^	38.40 ± 3.14 ^bc^	43.40 ± 0.82 ^bc^	38.00 ± 1.97 ^b^	47.05 ± 1.08 ^bc^
	120	47.20 ± 1.58 ^bc^	32.60 ± 2.02 ^bc^	46.00 ± 1.00 ^bc^	41.80 ± 0.95 ^bc^	50.60 ± 0.53 ^bc^
	150	56.60 ± 1.41 ^bc^	30.40 ± 2.10 ^bc^	54.30 ± 1.39 ^bc^	46.80 ± 1.44 ^b^	56.83 ± 0.74 ^bc^
	180	70.60 ± 1.92 ^bc^	20.50 ± 1.71 ^bc^	73.60 ± 1.06 ^bc^	55.90 ± 1.27 ^b^	69.90 ± 0.91 ^bc^
	0	66.40 ± 1.42	32.50 ± 1.76	63.50 ± 0.93	19.20 ± 0.80	54.15 ± 0.62
	30	30.80 ± 1.78	70.10 ± 2.12 ^a^	30.80 ± 1.65	20.60 ± 0.69 ^a^	28.03 ± 0.85 ^a^
	60	28.50 ± 1.66 ^a^	65.90 ± 1.57 ^a^	27.80 ± 1.80 ^ac^	22.00 ± 1.09 ^ac^	28.10 ± 0.58 ^ac^
**GS**	90	27.30 ± 1.95 ^ac^	63.70 ± 1.92 ^ac^	31.00 ± 1.69 ^a^	24.40 ± 1.75 ^ac^	29.75 ± 0.93 ^ac^
	120	29.80 ± 1.70 ^ac^	58.60 ± 1.99 ^ac^	35.60 ± 0.83 ^a^	33.30 ± 1.12 ^a^	35.03 ± 0.77 ^ac^
	150	34.30 ± 1.16 ^ac^	46.00 ± 1.74 ^ac^	43.90 ± 1.16 ^a^	36.60 ± 0.90 ^ac^	40.20 ± 0.42 ^ac^
	180	62.30 ± 1.51 ^a^	34.90 ± 1.47 ^ac^	58.80 ± 1.90 ^a^	45.00 ± 1.32 ^ac^	57.80 ± 0.64 ^ac^
	0	63.20 ± 1.30	32.80 ± 1.18	65.20 ± 1.24	21.40 ± 1.27	54.25 ± 0.59
	30	32.30 ± 1.50 ^a^	68.40 ± 1.38	29.60 ± 1.97	23.90 ± 1.30	29.35 ± 0.85 ^a^
	60	28.60 ± 2.03 ^ab^	60.60 ± 1.35	34.60 ± 1.48 ^ab^	25.50 ± 0.91 ^ab^	32.03 ± 0.68 ^ab^
**DI**	90	33.90 ± 1.38 ^ab^	54.80 ± 1.17 ^ab^	35.00 ± 1.22 ^a^	35.00 ± 1.02 ^b^	37.28 ± 0.64 ^ab^
	120	39.00 ± 1.13 ^ab^	45.90 ± 1.49 ^ab^	36.90 ± 1.17 ^a^	36.70 ± 1.27 ^a^	40.93 ± 0.58 ^ab^
	150	44.90 ± 0.99 ^ab^	38.30 ± 2.39 ^ab^	47.40 ± 1.97 ^a^	44.90 ± 1.50 ^b^	49.73 ± 0.63 ^ab^
	180	60.60 ± 2.02 ^a^	27.00 ± 1.71 ^ab^	61.90 ± 2.36 ^a^	52.00 ± 1.21 ^b^	61.88 ± 076 ^ab^

Treatment groups were defined as a standard nutritional supplement not specifically for people with diabetes (ET); resistant maltodextrin and sucromalt supplement (GS); isomaltulose and resistant starch supplement (DI). GIP: glucose-dependent insulinotropic polypeptide; GLP-1: glucagon-like peptide 1; SEM: standard error of the mean. ^a^
*p* < 0.05 versus ET. ^b^
*p* < 0.05 versus GS. ^c^
*p <* 0.05 versus DI. SA: Subjective appetite.

**Table 6 nutrients-11-01477-t006:** Coefficient correlations between AUC values of glycaemia, EIAP and subjective measurements of appetite according to consumption tests.

	Hunger		Fullness		Desire to Eat		Prospective Food Consumption		Subjective Appetite	
**ET**	***r***	***p***	***r***	***p***	***r***	***p***	***r***	***p***	***r***	***p***
Glycaemia	0.060	0.868	−0.687	0.028	0.217	0.547	−0.025	0.945	0.659	0.038
Insulin	−0.215	0.552	0.713	0.021	−0.046	0.900	0.362	0.304	−0.437	0.321
GLP-1	−0.133	0.714	−0.756	0.011	0.392	0.262	−0.543	0.105	0.321	0.540
GIP	0.219	0.544	−0.082	0.821	0.486	0.155	0.399	0.253	0.540	0.107
**DI**	***r***	***p***	***r***	***p***	***r***	***p***	***r***	***p***	***r***	***p***
Glycaemia	0.004	0.992	−0.226	0.530	−0.069	0.849	0.173	0.633	0.357	0.311
Insulin	0.190	0.599	−0.163	0.652	−0.455	0.187	−0.196	0.587	−0.254	0.479
GLP-1	0.483	0.158	0.140	0.700	0.158	0.662	−0.407	0.243	0.098	0.787
GIP	0.294	0.410	0.099	0.785	−0.615	0.058	0.069	0.850	−0.540	0.107
**GS**	***r***	***p***	***r***	***p***	***r***	***p***	***r***	***p***	***r***	***p***
Glycaemia	−0.192	0.595	0.019	0.958	−0.152	0.674	0.128	0.726	−0.154	0.672
Insulin	0.466	0.175	−0.020	0.957	0.217	0.548	−0.175	0.628	0.344	0.330
GLP-1	0.308	0.386	0.072	0.843	0.436	0.208	−0.421	0.226	0.191	0.596
GIP	0.231	0.521	−0.046	0.900	−0.135	0.710	−0.076	0.834	0.086	0.813

Treatment groups: (ET) standard nutritional supplement not specifically for diabetes patients; (DI) isomaltulose and resistant starch supplement; (GS) resistant maltodextrin and sucromalt supplement. GIP: glucose-dependent insulinotropic polypeptide; GLP-1: glucagon-like peptide 1; SEM: standard error of the mean. The values presented correspond to *r* coefficients and *p*-value for all subject correlations between subjective perceptions of appetite and concentrations hormones according to the treatment group. *p*-values were significant when *p* < 0.05.

**Table 7 nutrients-11-01477-t007:** Glycaemic index and glycaemic load according to consumption tests.

Treatment Groups	Mean ± SEM
**Glycaemic Index (GI)**	
ET	56.40 ± 0.43 ^bc^
DI	51.44 ± 0.60 ^ab^
GS	47.59 ± 0.49 ^ac^
**Glycaemic Load (GL)**	
ET	23.69 ± 0.18 ^bc^
DI	12.04 ± 0.14 ^ac^
GS	11.42 ± 0.12 ^ab^

Treatment groups: (ET) standard nutritional supplement not specifically for people with diabetes; (DI) isomaltulose and resistant starch supplement; (GS) resistant maltodextrin and sucromalt supplement. SEM: Standard error of the mean. There were no significant differences between sexes. ANOVA and post-hoc Tukey HSD for intragroup comparisons; *p*-value was significant when *p* < 0.05. ^a^
*p* < 0.001 versus ET. ^b^
*p* < 0.001 versus GS. ^c^
*p* < 0.001 versus DI.

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
