# Peer review of "Effect of Oral Nutritional Supplements with Sucromalt and Isomaltulose versus Standard Formula on Glycaemic Index, Entero-Insular Axis Peptides and Subjective Appetite in Patients with Type 2 Diabetes: A Randomised Cross-Over Study"

_nutrients, 2019, doi:10.3390/nu11071477_

Reviewer 1 Report

This study by Angrarita et studied the impact of oral nutritional supplement which contain complex carbohydrates in patients with type 2 diabetes. This is very important study and was conceived, designed, performed very carefully. The methods, inclusion and exclusion criteria and measurements were very well explained in the text. This study has significant finding; however, study needs to have more subjects along with healthy subjects.  Figures needs significant improvement along with some of the data is missing which should be included. I agree more or less with their finding, but authors need to address the following issues.

1)   Subject size is very small; therefore, this study needs to enroll more subjects and authors needs to present data on the basis of sex if possible. Heathy individuals should have been recruited for the study which would have strengthened this study significantly.

2)   Authors must include sucrose group to the study to show how it is compared to the three supplements tested. Actually, this comment was written before I started looking at the figures which in some graphs show GB group but why it is excluded from all the figures and tables.

3)   How many individuals were in one group and what is the ratio of males to females for each group? Since total subject size is pretty small and one group mainly comprises of males or females that could skew the results significantly. Authors needs to mention that in the figures and tables.

4)   In figure 1 key should be either on all the bar graphs or at least on the top graph (which is present in figure 2) and should be mentioned in the legend that same color scheme was used for all the graphs. They should also be labelled as A, B, C, D etc. in the figure which should be written clearly in the legend. None of the bar diagrams contains significance *. How is data presented is it Mean+SD or SE?

5)   Hunger and fullness are very subjective and hard to determine specially during controlled setting. There is no specific test and Therefore, although it is a finding but too much emphasis should not be given to these results.  Authors should concentrate on emphasis on glycemic control, GLP-1 findings more and also in discussion the limitations of these subjective tests should be mentioned.

Author Response

1) Subject size is very small; therefore, this study needs to enroll more subjects and authors needs to present data on the basis of sex if possible. Healthy individuals should have been recruited for the study which would have strengthened this study significantly.

A: Since postprandial glycemia and glycemic index (GI) was our primary outcome, this study was powered to detect the difference between the glucose area under the curve after consuming threes oral nutritional supplements (effect size = 0.79) [1,2,3,4]. Based on our calculation, at least 15 participants were needed to detect this effect size at 80% statistical power using a crossover study design [3,5]. Assuming an attrition rate of 30%, 23 participants were recruited in this study.

This paragraph has been included in the text, in the methods section corresponding to “Population, sample size, and patient’s selection,” with the objective of widening the description of the sample size.

Diverse studies with similar design and objectives have presented calculations for sample size similar to ours.[6,7,8]. However, we have underlined sample size precisely as one of the limitations of our work in order to study subjective appetite. One of the first investigations that analyzed the reproducibility, power and validity of the visual analogic scale (VAS), demonstrated that using a paired design and a study power of 0.8, a difference of 10 mm on fasting and 5 mm on mean 4.5 h ratings can be detected with 18 subjects [9]. The best correlations of validity were found between 4.5 h mean VAS of the appetite parameters. [9].  When using desires to eat specific types of food or an unpaired design, more subjects are needed due to considerable variation. In our study, appetite was assessed twice in the same subject before and after supplement intake on two different occasions for three hours.

We are conscious that after a desertion of almost 30%, our study simple did not reach an n of 18 subjects (only 16). However, in spite of the latter, research with relevant contributions specifically on the parameters of this variable, have also worked with small sample sizes, including 12-14 subjects[7-10]. On the order hand, for the determination glycaemic index, the inclusion of ten subjects provides a reasonable degree of power and precision for most purposes of measuring GI.[1-4]. We would like to underline that according to this design, each supplement, as well as the reference food (glucose solution) was tested in two different sessions for each subject. This supplementation decreases the margin of error in the IG values.[3].  

In another context, in our study design we did not consider the inclusion of healthy subjects, in light of that the literature has reported results from the principal variable (glycaemic index) from the three supplements in this group. However, we do confer that the criteria to be a larger contribution would have been a correlation of IG, incretins and subjective appetite in healthy subjects compared with subjects with diabetes, which we plan to do in future research.

 Moreover, we had not considered presenting the data differentiated by sex, given that the results of this variable did not produce any statistically significant findings. However we did decide to show the mean and SEM of the subjects’ characteristics based on sex.

2)   Authors must include sucrose group to the study to show how it is compared to the three supplements tested. Actually, this comment was written before I started looking at the figures which in some graphs show GB group but why it is excluded from all the figures and tables

A:  The glucose solution (GB) was administered solely to determine glycaemic index and insulin profile, as a product of the reference food, as recommended for this type of study design to determine glycaemic index. It is for this reason that it was not administered for the analysis of the rest of the hormone concentrations (GLP-1/GIP). Additionally, incretin behaviour after glucose consumption in diabetics has been reported in the literature, and therefore was not the objective of our study.

On the other hand, a second objective was to compare different parameters of subjective appetite after consuming the three nutritional supplements, it is only considered a reference product for determining the glycaemic profile. It was not administered for the rest of the hormonal determinations, nor for subjective appetite, given that it was not within the purpose that sustains the theory of our study.

3)   How many individuals were in one group and what is the ratio of males to females for each group? Since total subject size is pretty small and one group mainly comprises of males or females that could skew the results significantly. Authors needs to mention that in the figures and tables

A: The three nutritional supplements and the glucose solution (GB) were given to all of the participants (16 subjects, men and women) in a randomized way on two different occasions, as described in the study design portion of the study. Therefore, the sample size was the same for each treatment and active comparator, considering that they were given 3 treatments (nutritional supplements) and 1 active comparator or reference food (glucose solution). On the other hand, just as the Reviewer has suggested, we have indicated in the text, at the beginning of the results, the M/F proportion of the sample.

4) In figure 1 key should be either on all the bar graphs or at least on the top graph (which is present in figure 2) and should be mentioned in the legend that same color scheme was used for all the graphs. They should also be labelled as A, B, C, D etc. in the figure which should be written clearly in the legend. None of the bar diagrams contains significance *. How is data presented is it Mean+SD or SE?

A: We have made the respective corrections to the figure legend about the significance of the AUC value (bar diagram), indicated at the end of the legend (hormone levels as well as the parameters of subjective appetite). This data is presented as the mean + SEM (indicated in the figure legend).

5)  Hunger and fullness are very subjective and hard to determine specially during controlled setting. There is no specific test and Therefore, although it is a finding but too much emphasis should not be given to these results.  Authors should concentrate on emphasis on glycemic control, GLP-1 findings more and also in discussion the limitations of these subjective tests should be mentioned. 

A: We partially agree with the Reviewer about the opinion that hunger and fullness are very subjective and hard to determine especially during controlled settings. However, the tests applied in this research called visual analogic scales (VAS) have been used in clinical environments and research in orter to continually control an array of subjective sensations that are difficult to control such as depression, pain and appetite. In the last years, the use of VAS has progressed, and while some research groups use only classification from one component, others use a score composed of various components to calculate a general appetite score or appetite ratings, just as we did in our study. A series of revisions and key experimental studies have used VAS to measure appetite and have demonstrated a high grade of reproducibility [9-11], [12, 13, 14]. Different experimental studies have used the procedure of laboratory tests [15] in order to confirm the validity and reliability of VAS as a measurement of motivation to eat [16]. However, the specific measured parameters must be taken in mind, its sensitivity and study power. For these reasons we have mentioned within the limitations of the study the sample size for these type of subjective tests.

Proofreading and English language editing was done by a professional editing service. All changes to the manuscript are marked in red, as well as tracked changes used to show the revisions. Thank you for considering our study in your

References

Hofman Z, K.H. The Glycemic Index of standard and      diabetes-specific enteral formulas. Asia      Pac J Clin Nutr. 2006, 15,      412–7.

Atkinson, F.S.; Foster-Powell, K.; Brand-Miller,      J.C. International Tables of Glycemic Index and Glycemic Load Values:      2008. Diabetes Care 2008, 31, 2281–2283.

3.      Brouns F, W.TM. Glycaemic index methodology. Nutr Res Rev. 2005, 18, 145–71.

4.      Nilsson M, Stenberg M, Frid AH, Holst JJ, Björck IM (2004) Glycemia and insulinemia in healthy subjects after lactose-equivalent meals of milk and other food proteins: the role of plasma amino acids and incretins. Am J Clin Nutr 80(5):1246–1253

5.      Tan SY1, Siow PC1, Peh E1, Henry CJ2,3,4. Influence of rice, pea and oat proteins in attenuating glycemic response of sugar-sweetened beverages. Eur J Nutr. 2018 Dec; 57(8):2795-2803. doi: 10.1007/s00394-017-1547-3.

6.      Giezenaar, C.; Trahair, L.G.; Luscombe-Marsh, N.D.; Hausken, T.; Standfield, S.; Jones, K.L.; Lange, K.; Horowitz, M.; Chapman, I.; Soenen, S. Effects of randomized whey-protein loads on energy intake, appetite, gastric emptying, and plasma gut-hormone concentrations in older men and women. Am. J. Clin. Nutr. 2017, ajcn154377.

7.      Bornet, F.R.J.; Jardy-Gennetier, A.-E.; Jacquet, N.; Stowell, J. Glycaemic response to foods: Impact on satiety and long-term weight regulation. Appetite 2007, 49, 535–553.

8.      Giezenaar C, Luscombe-Marsh ND, Hutchison AT, Lange K, Hausken T, Jones K, Horowitz M, Chapman , Soenen S. Effects of Substitution, and Adding of Carbohydrate and Fat to Whey-Protein on Energy Intake, Appetite, Gastric Emptying, Glucose, Insulin, Ghrelin, CCK and GLP-1 in Healthy Older Men-A Randomized Controlled Trial. Nutrients. 2018 Jan 23;10 (2). pii: E113. doi: 10.3390/nu10020113.

9.       Flint A, A.A. Glycemic and insulinemic responses as determinants of appetite in humans. Am J Clin Nutr. 2006, 84, 1365–73.

10.  Giezenaar C, van der Burgh Y, Lange K3, Hatzinikolas S, Hausken T, Jones KL, Horowitz M, Chapman I, Soenen S Effects of Substitution, and Adding of Carbohydrate and Fat to Whey-Protein on Energy Intake, Appetite, Gastric Emptying, Glucose, Insulin, Ghrelin, CCK and GLP-1 in Healthy Older Men-A Randomized Controlled Trial.Nutrients. 2018 Jan 23;10(2). pii: E113. doi: 10.3390/nu10020113.

         11.Gibbons C1Hopkins M2Beaulieu K3Oustric P3Blundell JE3. Curr Obes Rep. Issues in      Measuring and Interpreting Human Appetite (Satiety/Satiation) and Its Contribution to Obesity. 2019 Jun;8(2):77-87. doi: 10.1007/s13679-019-00340-6.

12.    Stubbs R, et al. The use of visual analogue scales to assess motivation to eat in human subjects: a review of their reliability and validity with an evaluation of new hand-held computerized systems for temporal tracking of appetite ratings. Br J Nutr. 2000;84(04): 405–15.

13.    Gwaltney C, Shields A, Shiffman S. Equivalence of electronic and paper-and-pencil administration of patient-reported outcome measures: a meta-analytic review. Value Health. 2008;11(2):322–33

14.  Elehazara Rubio-Martín ,Eva García-Escobar , Maria-Soledad Ruiz de Adana Fuensanta Lima-Rubio , Laura Peláez , Angel-María Caracuel 1Francisco-Javier Bermúdez-Silva and et al .Comparison of the Effects of Goat Dairy and Cow Dairy Based Breakfasts on Satiety, Appetite Hormones, and Metabolic Profile Nutrients 2017, 9, 877; doi:10.3390/nu9080877

Flint A, Gregersen NT, Gluud LL, Møller BK, Raben A, Tetens I, et al. Associations between postprandial insulin and blood glucose responses, appetite sensations and energy intake in normal weight and overweight individuals: a meta-analysis of test meal studies. Br J Nutr. 2007;98(01):17–25.

16.Flint A1, Raben A, Blundell JE, Astrup A. Reproducibility, power and validity of visual analogue scales in assessment of appetite sensations in single test meal studies. Int J Obes Relat Metab Disord. 2000 Jan;24(1):38-48.

Reviewer 2 Report

Reviewer 1 comment´s:

This manuscript talks about the study of the effect of oral nutritional supplements with sucromalt and isomaltulose versus standard formula on glycaemic index, entero-insular axis peptides and subjective appetite in patients with type diabetes. Concretely, the authors compared postprandial effects of oral diabetes-specific nutritional supplements with (isomaltulose and sucromalt) versus standard formula on glycaemic index, insulin, glucose-dependent insulinotropic polypeptide, glucagon-like peptide 1 and subjective appetite in 16 individuals with Diabetus type 2 by randomised, double-blind, cross-over study.

The comments are described below:

Reviewer 1 comment´s:

-line 101: “…disease, and patients with cancer [27,28]”.

Please, put the final point that indicate the end of line (.)

-Line 109: “Moreover, a study by Pfeiffer et al. [32] …”

Please, put the reference number near author how indicated in author’s guidelines of journal.

-Line 115: “…selection of dietary carbohydrates. [32]”

Please, put the final point that indicate the end of line (.): “…selection of dietary carbohydrates [32].”

-Line 116: “Several studies underwent in DM-2 subjects…”

-Line 140: “ type 2 DM”

Please, put the correct form if you talk about of DM2 subjects, according your abbreviations.

2

-Line 144:” Patients with DM1,…”

Please, insert in your abbreviations DM1, if you consider patients with Diabetes Mellitus (DM) type 1, because there isn´t this word.

-Line 173,175,178: “100 mL”…

-Line 183,Table 1: Macronutrient composition of the oral nutritional supplements per 100 ml

Please, the numbers and their units must be united in the sentences, and following the same.

Please, Check all manuscript to verify these incorrect forms and homogenize the text.

-Line 732: Abbreviations: “AUCG”

It is not in italics, please, correct it.

Reviewer 1 comment´s:

References:

-I propose to review all references following the indications to the authors' guidelines.

References should be described as follows, depending on the type of work:

Journal Articles: Author 1, A.B.; Author 2, C.D. Title of the article. Abbreviated Journal Name Year, Volume, page range, DOI. Available online: URL (accessed on Day Month Year).

*i.e. Lines 755,794,806, 809… Please put the correct form: Abbreviated Journal, without points

Websites: Title of Site. Available online: URL (accessed on Day Month Year).

Author Response

Thank you for this helpful review. We have taken into consideration all of the reviewer's comments, and we answer each one below in red text.  

Reviewer 1 comment´s:

This manuscript talks about the study of the effect of oral nutritional supplements with sucromalt and isomaltulose versus standard formula on glycaemic index, entero-insular axis peptides and subjective appetite in patients with type diabetes. Concretely, the authors compared postprandial effects of oral diabetes-specific nutritional supplements with (isomaltulose and sucromalt) versus standard formula on glycaemic index, insulin, glucose-dependent insulinotropic polypeptide, glucagon-like peptide 1 and subjective appetite in 16 individuals with Diabetus type 2 by randomised, double-blind, cross-over study.

The comments are described below:

Reviewer 1 comment´s:

-line 101: “…disease, and patients with cancer [27,28]”.

Please, put the final point that indicate the end of line (.)

We have fixed this formatting error and put the point on the end of the line, per the reviewer's suggestion. 

-Line 109: “Moreover, a study by Pfeiffer et al. [32] …”

Please, put the reference number near author how indicated in author’s guidelines of journal.

We have put the reference number near the author, as indicated in the author guidelines.

-Line 115: “…selection of dietary carbohydrates. [32]”

Please, put the final point that indicate the end of line (.): “…selection of dietary carbohydrates [32].”

We have fixed this formatting error and put the point on the end of the line, per the reviewer's suggestion. 

-Line 116: “Several studies underwent in DM-2 subjects…”

We have taken out the word "underwent" to make this line more readable. We have also homogenized the abbreviation to read DM2 instead of DM-2. 

-Line 140: “ type 2 DM”

Please, put the correct form if you talk about of DM2 subjects, according your abbreviations.

2

We fixed this abbreviation per the reviewer's comment. 

-Line 144:” Patients with DM1,…”

Please, insert in your abbreviations DM1, if you consider patients with Diabetes Mellitus (DM) type 1, because there isn´t this word.

We inserted the words "diabetes mellitus type 1" to clarify the abbreviation, thank you. 

-Line 173,175,178: “100 mL”…

-Line 183,Table 1: Macronutrient composition of the oral nutritional supplements per 100 ml

Please, the numbers and their units must be united in the sentences, and following the same.

We homogenized the abbreviation mL (instead of ml) and have united the numbers and the units, according to the reviewer's comment. 

Please, Check all manuscript to verify these incorrect forms and homogenize the text.

We verified and corrected all of these abbreviated forms and homogenized the text according to the reviewer's suggestion. 

-Line 732: Abbreviations: “AUCG”

It is not in italics, please, correct it.

We think the reviewer meant "bold text" not italic text, and we have bolded the text to match the rest of the abbreviation list. 

Reviewer 1 comment´s:

References:

-I propose to review all references following the indications to the authors' guidelines.

References should be described as follows, depending on the type of work:

Journal Articles: Author 1, A.B.; Author 2, C.D. Title of the article. Abbreviated Journal Name Year, Volume, page range, DOI. Available online: URL (accessed on Day Month Year).

*i.e. Lines 755,794,806, 809… Please put the correct form: Abbreviated Journal, without points

Websites: Title of Site. Available online: URL (accessed on Day Month Year).

We have meticulously reviewed and formatted the references, per the reviewer's request. However, we would like to point out that per the  Full MDPI Reference List and Citations Style Guide (https://mdpi-res.com/data/mdpi_references_guide_v5.pdf) as well as the ISO4 standard reference guidelines, in fact, abbreviated titles do appear with their respective periods. Therefore, we have left the "points" as described by the reviewer, and have homogenized the references according to the Reference List and Style Guide and the ISO4 Standard.